# Ocular-following responses in school-age children

**Aleksandar Miladinović**[1,2]*, **Christian Quaia**[3], **Miloš Ajčević**[2], **Laura Diplotti**[1], **Bruce G. Cumming**[3], **Stefano Pensiero**[1], **Agostino Accardo**[2]

**1** Department of Ophthalmology, Institute for Maternal and Child Health-IRCCS Burlo Garofolo, Trieste, Italy, **2** Department of Engineering and Architecture, University of Trieste, Trieste, Italy, **3** Laboratory of Sensorimotor Research, National Eye Institute, National Institutes of Health, Department of Health and Human Services, Bethesda, MD, United States of America

* aleksandar.miladinovic@burlo.trieste.it

## Abstract

Ocular following eye movements have provided insights into how the visual system of humans and monkeys processes motion. Recently, it has been shown that they also reliably reveal stereoanomalies, and, thus, might have clinical applications. Their translation from research to clinical setting has however been hindered by their small size, which makes them difficult to record, and by a lack of data about their properties in sizable populations. Notably, they have so far only been recorded in adults. We recorded ocular following responses (OFRs)–defined as the change in eye position in the 80–160 ms time window following the motion onset of a large textured stimulus–in 14 school-age children (6 to 13 years old, 9 males and 5 females), under recording conditions that closely mimic a clinical setting. The OFRs were acquired non-invasively by a custom developed high-resolution video-oculography system, described in this study. With the developed system we were able to non-invasively detect OFRs in all children in short recording sessions. Across subjects, we observed a large variability in the magnitude of the movements (by a factor of 4); OFR magnitude was however not correlated with age. A power analysis indicates that even considerably smaller movements could be detected. We conclude that the ocular following system is well developed by age six, and OFRs can be recorded non-invasively in young children in a clinical setting.

## Introduction

The sudden motion, in an unpredictable direction, of a large textured pattern in the visual field induces reflexive eye movements at ultra-short latencies (around 50 ms in monkeys, 70 ms in humans). Such eye movements are called ocular following [1–6]. They represent the initial component of the optokinetic nystagmus response, which supports the translational vestibulo-ocular reflex system in the stabilization of gaze [5]; the recent discovery that they can be induced, albeit at longer latencies, even by relatively small stimuli [7, 8] suggests that they might also contribute to the initial phase of smooth pursuit eye movements. Because they are

**Data Availability Statement:** Raw data cannot be shared publicly due to privacy. The data contain the captures of the eyes of the children participated in the study. According to the policy of the IRCCS Burlo Garofolo, Trieste, such data are categorised

as sensitive as can potentially reveal identity of the study participants. However, the intermediate processed data is available for download from https://github.com/miladinovic/ofr/tree/main/ofr_rds_responses

**Funding:** This study was funded by the Italian Ministry of Health, Grant/Award Number: 2764554 Ricerca Corrente 2021; CQ and BGC were supported by the Intramural Research Program of the National Eye Institute. The funders had no role in study design, data collection and analysis, decision to publish, or preparation of the manuscript.

**Competing interests:** The authors have declared that no competing interests exist.

typically measured over a very short time period (the so-called open-loop period, corresponding to the time window between the latency and twice the latency of the response, when visual reafference cannot affect its properties), their magnitude is very small (in adults, a stimulus drifting at 50°/s, which induces the strongest ocular following responses, will typically trigger an ocular following movement with a speed of around 2°/s, thus compensating for only 4% of the retinal slip, see Fig 1), making recording them difficult, and their functional role questionable. Nonetheless, their strong dependency on the size, contrast, spatial and temporal frequency content, and even inter-ocular correlation of the drifting stimulus, has made them a powerful tool to probe the computations underlying visual motion extraction in humans and non-human primates (and recently even in mice).

The small size of these movements has unfortunately limited their appeal, so that most ocular following studies have been produced by a handful of research groups. Measuring them is undoubtedly challenging, requiring the use of highly sensitive eye tracking systems (such as scleral search coils, dual-Purkinjie eye trackers, or modern video-based eye tracking systems) and averaging over many responses (often 100 trials or more for each stimulus condition tested), typically collected in long sessions over multiple days; many studies also take advantage of the significant enhancement in their size observed when they are triggered in the wake of a saccade (post-saccadic enhancement, [9]).

It has been recently demonstrated that ocular following responses are mediated by disparity-sensitive cortical neurons, and are thus sensitive to interocular correlations [10]. This property has then been used to demonstrate that these movements can be used, in a research setting using adult subjects instrumented with scleral search coils, to reliably diagnose stereoanomalies [11]. Of course, there is no shortage of clinical tests to diagnose stereoanomalies in adults, but these eye movements, thanks to their minimal requirements for cooperation from the subject, could potentially be valuable tools in children, especially pre-verbal ones, or in non-verbal subjects of any age.

Unfortunately, as a result of the obstacles outlined above, ocular following eye movements have never been recorded in anyone under 18 years of age, in a clinical setting (where a single, short, recording session is the standard), or even in any group of subjects large enough to be able to carry out the power analysis necessary to plan a clinical study in infant populations (the typical subject size is two in monkeys, and three to four in humans). Research has been so narrowly focused on how various manipulations of the visual inputs affect their relative size and latency, that little is known about their absolute size and its variability across subjects.

The goal of this study is to fill these gaps, and to demonstrate that it is feasible to use a non-invasive video-based eye tracking system, which we describe in detail in the Methods section (High-resolution video-oculography), to record ocular following eye movements in young children (6 to 13 years old), in a setting that is as close as possible to a clinical one.

## Methods

### Study population

Sixteen healthy children without visual defects other than mild myopia or mild hyperopia (± 1D) were enrolled at the Ophthalmology Department of the Institute for Maternal and Child Health–IRCCS Burlo Garofolo (Trieste, Italy). All subjects underwent a complete orthoptic and ophthalmological exam. Each child was first examined by an ophthalmologist to determine visual acuity through non-cycloplegic refraction; the parents of those that met our criteria were then approached to receive written informed consent; in those that agreed, a recording session was then performed; finally, the child was seen again by the ophthalmologist for a cycloplegic refraction test (and usually additional tests). We planned to subsequently

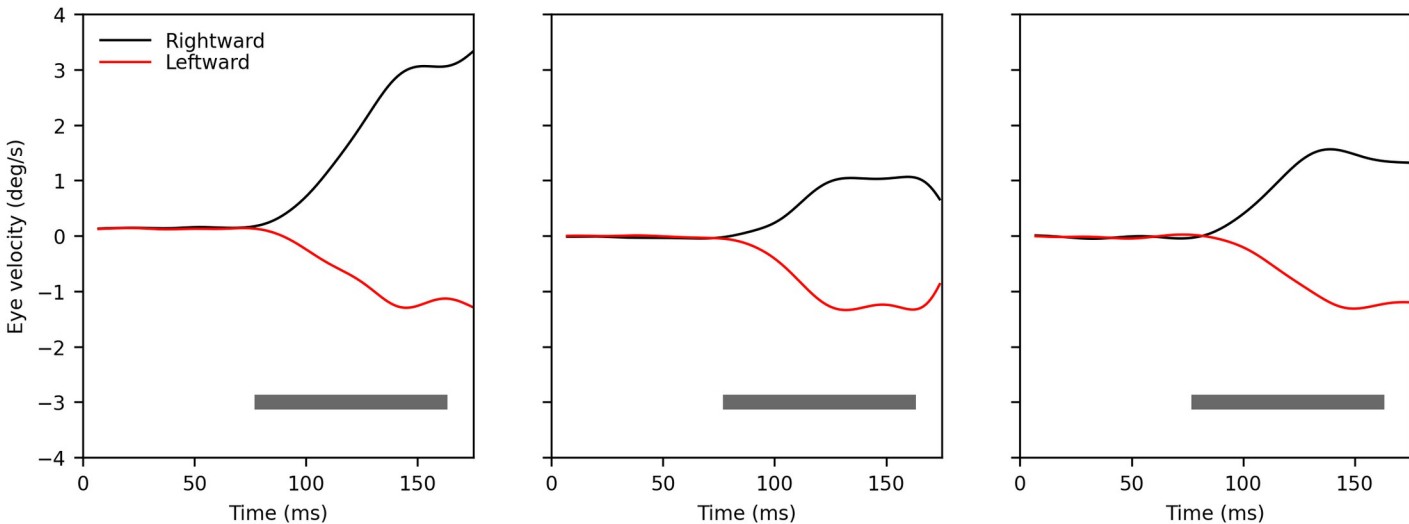

**Fig 1. Typical ocular following velocity traces induced by a large patch of moving random dots, similar, but not identical, to those used in the experiments described later.** In this case the dots were smaller, drifted at 36.5˚/s, and the aperture was square (25˚x25˚). Eye movements were recorded with a scleral search coil. Each panel shows data from a different subject, to highlight the commonly observed variability across subjects in both magnitude and dynamic evolution of the ocular following response. The gray bar represents the open-loop period, the time interval over which ocular following is typically quantified (in terms of either displacement or average velocity), which here was 80–160 ms.

exclude from the study, without analyzing their data, children with a significant difference between the two refraction tests, but there were none. Two subjects were excluded from the study due to a lack of cooperation. The mean age of included subjects (n = 14) was 9.71 ± 2.67 years (range 6–13 yo, 9 males, 5 females). The three subjects who were prescribed vision correction were instructed to wear glasses during the recordings.

## Ethics statement

Written informed consent was obtained from a parent (or legal guardian) of each participant. The parent was informed that the test was of no direct benefit to his/her own child, that participation was entirely voluntary and unrelated to the clinical care that the child was receiving at the eye clinic on that day, and that declining participation in no way affected the clinical care that the child would receive now or in the future at the hospital. The parent was shown the equipment used, and a few sample trials were run to illustrate the visual stimulation and what was expected of the child. The parent was also present throughout the testing session. The research was approved by the Institutional Review Board of the IRCCS Burlo Garofolo and adheres to the tenets of the Declaration of Helsinki.

## Behavioral paradigm

Children sat in a dimly illuminated room with their heads stabilized using chin and forehead padded supports and a loosely tied head band, facing a monitor (ASUS VG248QE, set at a resolution of 1920x1080 pixels, and at a vertical refresh rate of 144Hz), located 50 cm in front of the corneal vertex. The chair height was adjusted so that the eyes of the subject were aligned with the center of the screen. Vision was binocular (Fig 2A).

Ocular following eye movements were induced by presenting random-dot stimuli (RDS), drifting at high speed (50˚/s) either upward (UW), downward (DW), leftward (LW) or rightward (RW). Stimuli were created by subdividing the screen into a checkerboard, where each square had a side of 11 pixels (0.44˚). Each square was then randomly assigned either a high or

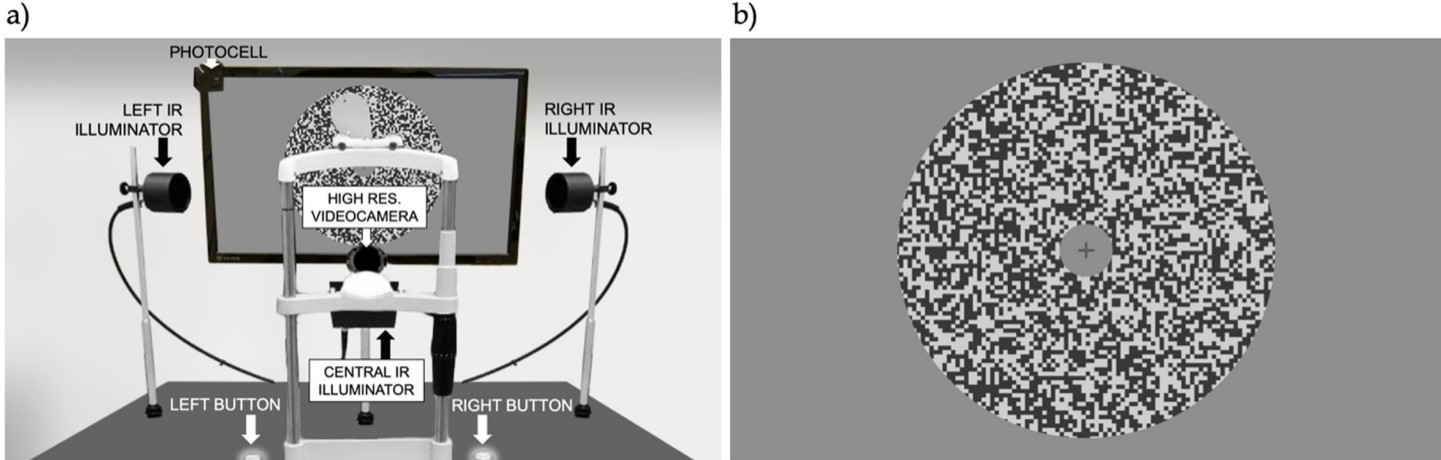

**Fig 2. A)** Our recording apparatus. **B)** Sample stimulus (a RDS, which once the fixation cross is extinguished starts drifting at high speed (50˚/s) either upward, downward, leftward or rightward for 200 msec).

a low luminance value. Stimuli had a mean luminance of 42.0 cd/m$^2$ and 80% Michelson contrast, and were presented within a circular aperture (28˚ diameter) centered on the screen, as if drifting behind it (Fig 2B). These stimulus parameters were selected to maximize the expected magnitude of the ocular following movements, based on previous tuning studies in adult subjects [6, 12–14].

The experiment consisted of 25 blocks, each composed of four randomly intermixed trials (one for each drifting direction of the stimulus). Each trial began with the screen filled with a mid-luminance (42.0 cd/m$^2$) gray, on which the first frame of the RDS was shown. A small cross was presented at the center of the screen, and subjects were instructed to fixate its center and avoid blinking. After 800 to 1100 ms the fixation cross disappeared, and the RDS visual stimulus started drifting (as if moving behind the fixed circular aperture); motion lasted approximately 200 ms. Subsequently, the screen was blanked (again at mid-luminance), signaling the end of the trial, and after a short inter-trial interval a new trial was started.

To ensure the engagement of the children, an inter-trial "game" was used. In a fraction of randomly selected trials ("game" trials), right after the last frame of the drifting stimulus, the face of an alien cartoon character was briefly flashed at the center of the screen. The alien was depicted in half-profile and could face left or right. The child was instructed to press one of two buttons to report the direction of the alien. After the response, an acoustic signal (different for correct and incorrect responses) was generated, and the next trial was started.

### High-resolution video-oculography

Because of their small size, ocular following responses have been traditionally recorded with scleral search coils. Example OFR eye velocity traces recorded with this method are shown in Fig 1. The responses elicited in three adult subjects by the same stimuli (a RDS like the one we use in this study, but with smaller dots and presented in a square aperture, drifting leftward or rightward at 36.5˚/s) are shown in the three panels. It can be readily appreciated that the dynamic properties of ocular following movements are idiosyncratic, that there can be directional asymmetries, and that the magnitude of the movements can be considerably different across subjects (the scale is the same in the three panels). To quantify the OFR magnitude, all studies report the displacement (or the average velocity) in a fixed time period relative to the onset of the stimulus, typically the open-loop period (gray bar, here 80–160 ms as in our

study), which covers the time between the latency to twice the latency. What is quantified is thus the difference between the position of the eye at these two time points, regardless of its trajectory in between.

Given the intrusive and uncomfortable nature of search coil recordings, video-based methods are however an almost obligated choice for recording in children. Although high-end commercial eye trackers have been used successfully to record ocular following [15], they typically require averaging over a large number of trials. We thus developed a custom acquisition system that uses a video camera to record images of the dominant eye during visual stimulation, and that trades off high temporal resolution for high spatial resolution.

Our custom acquisition system consists of a high-resolution camera (FLIR Grasshopper 3 GS3-U3-51S5M-C, C-Mount 50mm f/2.8 lens, with a resolution of 2448x2048 pixels) recording in the near-infrared spectral range (a Hoya R72, IR filter with cut-off wavelength of 720 nm was placed in front of the lens to block the visible spectrum). To ensure proper lighting, three custom-built infrared (IR) LED illuminators (one on each side of the subject and one in front and below the subject) were used (Fig 2A). On each trial, we acquired three frames: $t_0 = 0$ ms (fixation cross offset/motion onset), $t_1 = 80$ ms, corresponding to a typical ocular following latency in human adults to stimuli of the size and contrast used here [7, 8, 12, 14], and $t_2 = 160$ ms (end of the open-loop period of the movement, twice the latency). To minimize the IR power delivered to the eye the illuminators were switched on for only 2ms, synchronized with the acquisition of a frame by the camera (the camera exposure duration was set to 1ms). IR lighting was thus on for only 6 ms in each trial (the trial duration varied from approximately 1.3 s in "no-game" trials to 2.5–3 s in "game" trials), a very low duty cycle that delivered minimal infrared power to the eyes (in contrast to commercial eye tracker illuminators, which are on continuously). The camera shutter and the illuminators were triggered by an Arduino-based controller connected to a photocell placed in front of the top left corner of the monitor. On the frame on which the fixation point was turned off and the RDS started drifting, the luminance of the area of the screen under the photocell was increased, allowing the circuit to detect $t_0$, turn on the illuminators for 2 ms and send a trigger pulse to the camera shutter. 80 and 160 ms later the Arduino controller automatically repeated the process (without the need of a photocell trigger).

In our experiment, only the location of the dominant eye was measured, together with that of a marker (an IR-absorbing black circle on a small sticker) that was placed on the bridge of the nose of the subject (this location was chosen because it is least affected by changes in facial expression). The purpose of the marker was to track motion of the head, and thus compensate for it to infer the actual eye-in-head displacement (the measure we are interested in). We did not apply the commonly used corneal-reflex technique to compensate for head motion, because while it would have been sufficient in the present experiments, it would not be applicable to infant recordings, where a larger range of eye displacements is to be expected. Our solution is thus similar to the so-called "Remote option" of the SR Research EyeLink 1000 eye tracker system, although the placement of the marker is quite different in our system.

We calibrated our system using an artificial eye (as it is standard for all eye tracking devices), and by comparing its performance directly to that of search coils (by recording ocular following responses in two subjects while they wore an eye coil in the eye being recorded with the video camera). The RMS resolution of the system (SD of the measured size of small displacements– 15 μm, corresponding to approximately 0.08˚ - of the artificial eye) for the artificial eye was 0.01˚; the RMS resolution for a human eye (inferred from the amplitude of the change in eye position associated with ocular following eye movements in the 80–160 ms open loop period, measured simultaneously with search coils and our system) was 0.036˚. These values are approximately twice as large as those of a typical search coil system, and compare

favorably with the reported resolution of commercial eye tracking systems (although a direct comparison of search coils and a commercial eye tracker for OFRs has not been published). Note that a limiting factor of the reliability in-vivo is the shape of the pupil, which is not always perfectly elliptical, and whose location is not rigidly tied to the direction of the visual axis, introducing a hard limit on the spatial resolution of any pupil-based eye position determination [16, 17].

## Data analysis and curation

Eye displacements were measured off-line, based on the images acquired and stored on disk during the experiment. For each image, two small rectangular areas, one containing the head position marker and the other containing the pupil of the dominant eye, were found using an automated algorithm, following the manual selection of ROIs corresponding to the eye of interest and the head position marker in the first recorded image. Using the frame acquired at time $t_0$ as a reference, our algorithm then extracted with sub-pixel resolution the displacement of the head marker and pupil center at frames $t_1$ and $t_2$. If the algorithm failed to find the displacement of marker or/and eye, the trial was considered invalid. This occurred when the subject blinked or moved the head by a large distance. The displacements of the eye (eye-in-head) during the fixation epoch (frames $t_1$ vs $t_0$) and the movement epoch (frames $t_2$ vs $t_1$) were then computed by subtracting the head marker displacements (head-in-space) from the pupil displacements (eye-in-space). These measures were all computed in pixels and were there converted to degrees of visual angle based on the geometry of our recording systems and assuming an eye diameter of 22 mm (average for children in this age group [18]; note that all our subjects had only no or minor refractive errors, making this assumption appropriate), resulting in a conversion factor of 0.168˚/pixel. Note that small differences in this value would not impact our conclusions, which mostly rest on the ability to detect the responses. Because the eye diameter can be considerably smaller in younger children [19], a direct measurement of the axial length might be necessary if accurate measurements of the magnitude of the response were found to be desirable. Calibration procedures asking subjects to look at targets of known eccentricity, standard in video-based recordings, were not carried out, as they would unnecessarily increase testing time, and might be cumbersome in younger children.

Next, we applied a procedure to automatically exclude from the analysis trials in which fixation was poor (for example, because of the presence of saccades, microsaccades, or large head movements). During the fixation period, neither the head nor the eyes should move. During the movement period, the eyes should move but not the head. We thus considered the displacements of the head (in space) and the eye (in space and relative to the head) during the fixation period, and the displacement of the head (in space) during the movement period, and excluded from further analysis any trials in which any of these measures was deemed an outlier. For each of these measures, we sorted in ascending order its value across all trials (i.e., regardless of which stimulus was presented), found the middle 68th percentile of these values, and marked as outliers any trials in which the value for the measure was larger than three times the 68th percentile value. This resulted in excluding between 5% and 25% of the trials (15% on average; note that these were on top of those excluded at the image processing stage because of blinking).

For each subject, the mean and standard deviation (SD) of horizontal and vertical eye displacements in the fixation and movement epochs were then calculated, separately for each stimulus direction. Tests of normality (Jarque-Bera and Shapiro-Wilk) were also applied to each condition, and 20% of conditions failed the normality tests. Because there is no consensus on whether corrections for multiple-comparisons should be applied to these tests (in which

case no conditions failed the normality tests), we decided to use both parametric and non-parametric tests for all our statistical analyses, and present the results of both.

The horizontal displacements induced by horizontal motion (rightward vs leftward), and the vertical displacements induced by vertical motion (upward vs downward), were compared using Student's two-tailed unpaired t-test and Mann-Whitney U test, with a 0.05 level for significance (separately for the fixation and movement epochs). The effect size during the movement period was then evaluated using Cohen's $d$. If we indicate with $D_R$ and $s_R$ the mean and SD of the horizontal displacement in $N_R$ rightward motion trials, and with $D_L$ and $s_L$ the mean and SD of the horizontal displacement in $N_L$ leftward motion trials, this is defined as

$$d_H = \frac{|D_R - D_L|}{\sqrt{\frac{(N_R-1)s_R^2 + (N_L-1)s_L^2}{N_R+N_L-2}}}$$

$d_V$ can be similarly defined for vertical displacements. A power analysis was then carried out, again separately for horizontal displacements induced by horizontal motion and vertical displacements induces by vertical motion, to identify the minimum effect size that, given the number of trials collected, could have been detected 80% of the times at the 0.05 significance level. The parametric analysis was based on the unpaired t-test, whereas the non-parametric one was based on bootstrap and the Mann-Whitney U test.

Finally, we computed the difference (mean and SD) between the horizontal displacements induced by rightward and leftward motion, and the difference between the vertical displacements induced by upward and downward motion. These are usually referred to as the Ocular Following Response (OFR), although sometimes the term ocular following amplitude is also used. Horizontal and vertical OFRs across subjects were compared using a paired t-test and Wilcoxon signed rank test, with a 0.05 significance level. The difference between horizontal and vertical OFR in individual subjects was evaluated using a bootstrap-based test. Correlations between the OFRs and Cohen's $d$s with age were evaluated computing Pearson's and Spearman's correlation coefficients.

All statistical analyses were performed in Python, using the freely available *statsmodels* and *scipy* packages. Most of the results of the statistical tests are reported in tables, and the others are reported in the Results section.

## Results

In Fig 3 we show scatter plots of horizontal and vertical eye displacements, separately for rightward and leftward drifting stimuli (top) and upward and downward drifting stimuli (bottom), recorded from one representative subject (Subject 10). In the left column the displacement during fixation ($t_1$ vs $t_0$, 0–80 ms from stimulus onset) is shown, whereas in the right column the displacement during the movement-epoch ($t_2$ vs $t_1$, 80–160 ms from stimulus onset, the so-called open-loop period in which the magnitude of ocular following movements is traditionally evaluated) is shown. Small dots indicate individual trials, whereas a larger dot indicates the mean (bars indicating ±1 standard-error-of-the-mean are also shown). The dots are color-coded according to the direction of motion of the stimulus. During the fixation epoch, displacements were small and unrelated to the direction of motion of the RDS. During the movement epoch, eye displacements were instead sizable (on average around 0.2˚-0.25˚ in each direction), and significantly different according to the direction of motion (horizontal: leftward vs rightward, vertical: upward vs downward).

In Fig 4 scatter plots of the average eye displacements for all subjects during the fixation (left) and movement (right) epochs are shown. In the fixation period, there was no significant

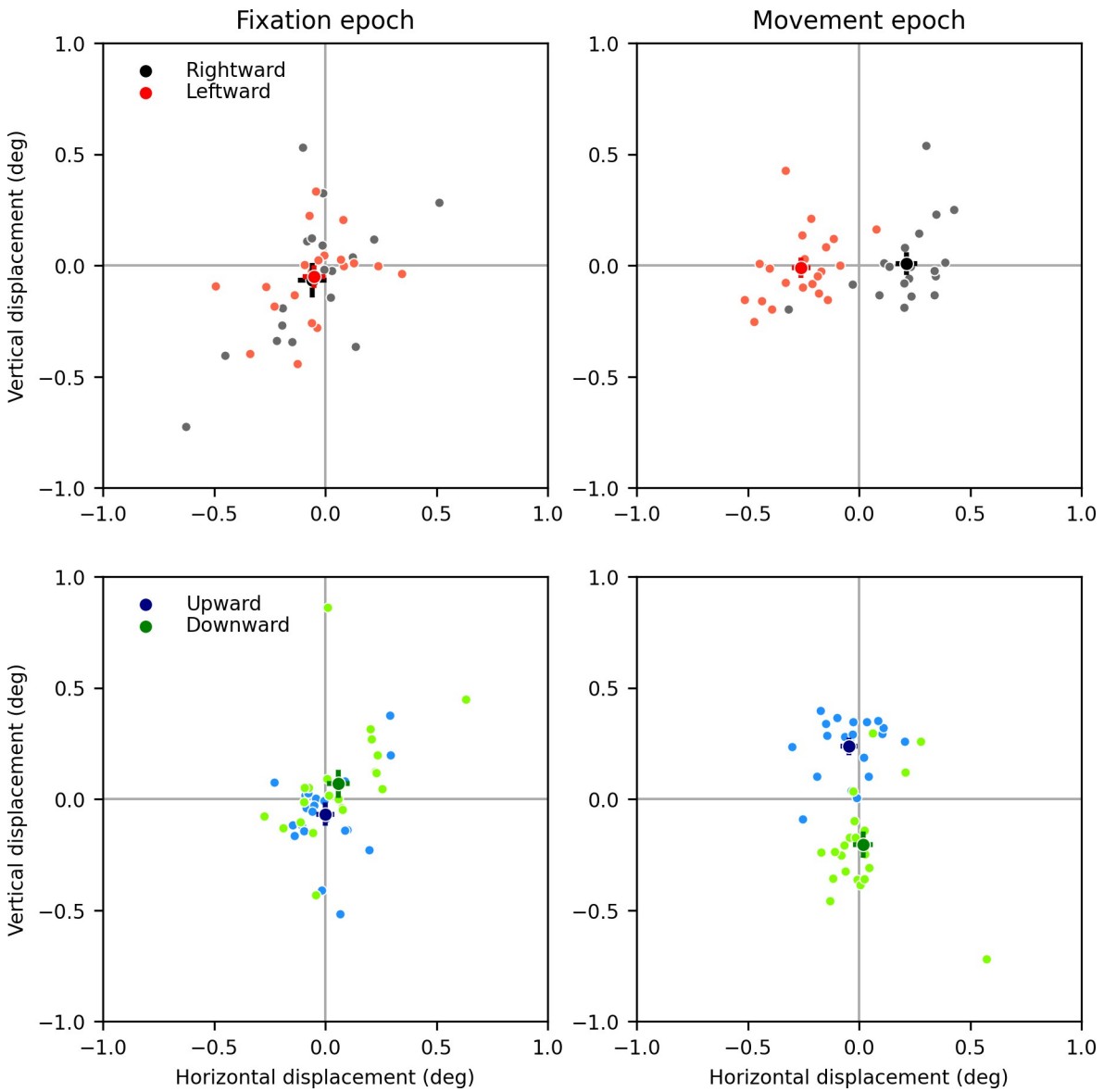

**Fig 3.** Eye displacements recorded in one representative subject during the fixation (left panels) and movement (right panels) epochs. The top panels show eye displacements induced by horizontally drifting stimuli, whereas the bottom panels show eye displacements induced by vertically drifting stimuli. Small symbols are individual trials, large symbols are means (with ±1 SEM bars).

displacement, either horizontally or vertically, in any of the subjects. In the movement epoch, all subjects showed significant differences between vertical displacements induced by RDS drifting vertically (upward vs downward), and between horizontal displacements to stimuli drifting horizontally (rightward vs leftward). Subject 11 showed a difference that was almost significant using a parametric test (p = 0.0513), but clearly significant with a non-parametric test (p = 0.0157). We thus consider that result significant as well. All other differences were highly significant with both parametric and non-parametric tests (see Tables 1 and 2). Numerical values for individual subjects are listed in Table 1 for horizontal motion, and in Table 2 for vertical motion.

We only measured eye displacement over two time intervals (0-80ms and 80-160ms), which may seem different from many studies of ocular following, which frequently show plots

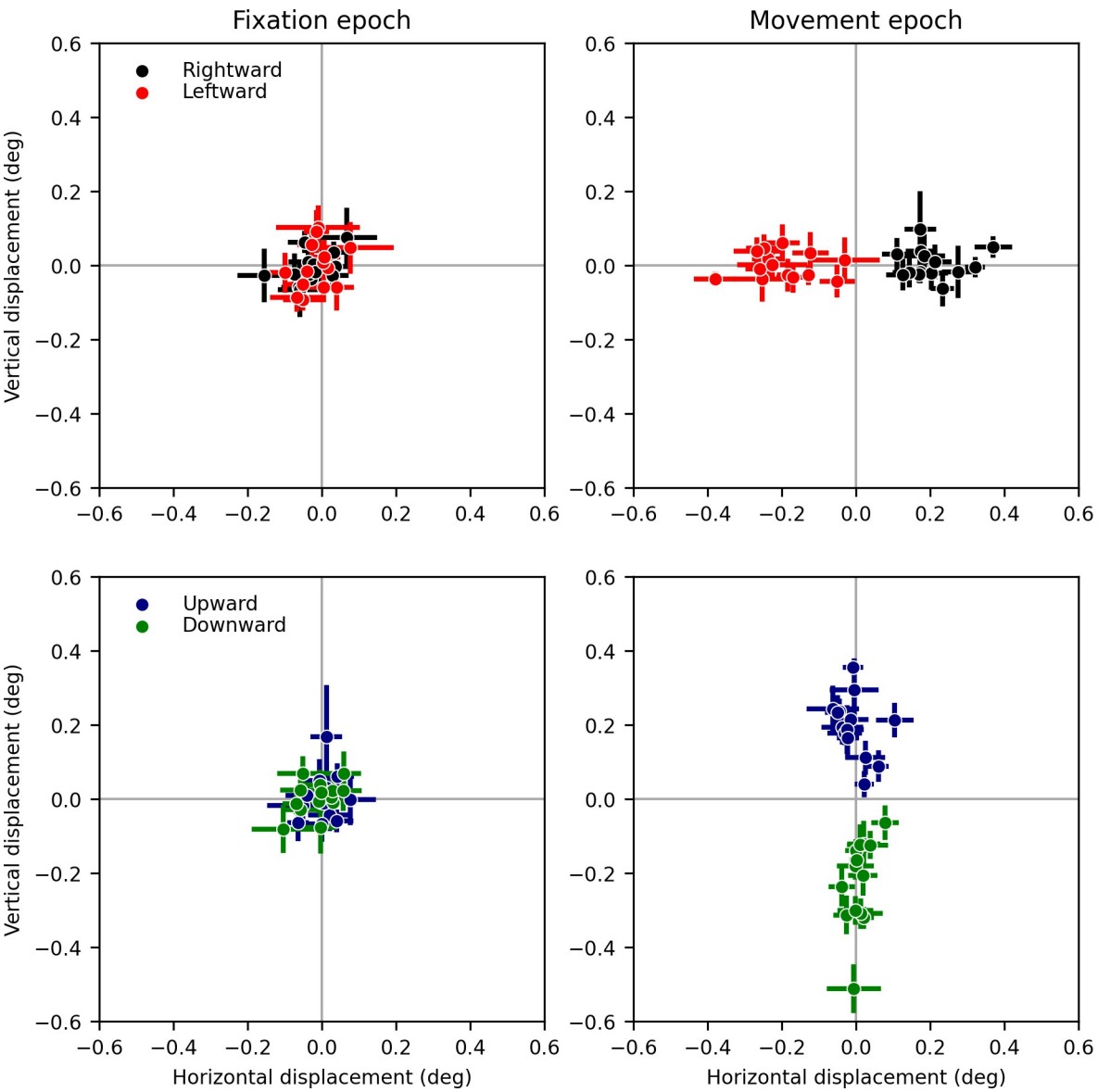

**Fig 4.** Average (and ±1 SEM bars) eye displacements recorded in our 14 subjects, during the fixation (left panels) and movement (right panels) epochs. The top panels show eye displacements induced by horizontally drifting stimuli, whereas the bottom panels show eye displacements induced by vertically drifting stimuli.

of instantaneous eye velocity (as in Fig 1). However, while these velocity traces are useful to estimate latencies, ocular following studies then rely on displacement measures over similar time intervals (shaded gray area in Fig 1) for their quantification, just like we did.

It is evident from Fig 3 that there is quite a lot of variability in the amplitude of individual ocular following responses to the same stimulus (as measured here). To directly estimate the impact of this variability on our ability to detect ocular following in children, we computed Cohen's *d*, a measure of effect size. We found that it was on average 2.75 for vertical motion and 2.45 for horizontal motion (see Tables for per-subject values). A power analysis indicated that, with the number of trials collected, the minimum effect size that could have been detected was around 1.00 (exact values in tables), indicating that we could detect considerably smaller

**Table 1. Responses to horizontally drifting images.**

| Subj | Age/Sex | Δx±SD(N) RW | Δx±SD(N)LW | p | p$_{NP}$ | d | min d | min d/d | min d$_{NP}$ | min d$_{NP}$/d | OFR_V±SD |
|------|---------|-------------|------------|---|-----|---|-------|---------|---------|-----------|----------|
| 1 | 13/M | 0.321±0.079 (15) | -0.379±0.192 (14) | <0.001 | <0.001 | 4.83 | 1.08 | 0.22 | 2.42 | 0.50 | 0.675±0.106 |
| 2 | 8/M | 0.173±0.128 (17) | -0.185±0.093 (17) | <0.001 | <0.001 | 3.20 | 0.99 | 0.31 | 1.00 | 0.31 | 0.211±0.279 |
| 3 | 8/F | 0.276±0.145 (15) | -0.253±0.251 (13) | <0.001 | <0.001 | 2.63 | 1.10 | 0.42 | 1.15 | 0.44 | 0.531±0.339 |
| 4 | 7/M | 0.202±0.164 (18) | -0.238±0.156 (19) | <0.001 | <0.001 | 2.75 | 0.95 | 0.34 | 0.96 | 0.35 | 0.503±0.219 |
| 5 | 12/M | 0.233±0.108 (19) | -0.248±0.176 (17) | <0.001 | <0.001 | 3.34 | 0.96 | 0.29 | 1.06 | 0.32 | 0.526±0.246 |
| 6 | 12/M | 0.110±0.165 (23) | -0.051±0.199 (23) | 0.005 | 0.002 | 0.88 | 0.84 | 0.96 | 0.80 | 0.91 | 0.316±0.164 |
| 7 | 13/F | 0.175±0.144 (18) | -0.170±0.175 (21) | <0.001 | <0.001 | 2.13 | 0.92 | 0.43 | 0.73 | 0.34 | 0.515±0.198 |
| 8 | 13/M | 0.184±0.229 (22) | -0.198±0.191 (24) | <0.001 | <0.001 | 1.82 | 0.85 | 0.46 | 0.59 | 0.32 | 0.369±0.317 |
| 9 | 12/M | 0.170±0.102 (22) | -0.128±0.057 (23) | <0.001 | <0.001 | 3.61 | 0.85 | 0.24 | 1.80 | 0.50 | 0.205±0.153 |
| 10 | 6/F | 0.213±0.166 (19) | -0.260±0.145 (21) | <0.001 | <0.001 | 3.04 | 0.91 | 0.30 | 0.78 | 0.25 | 0.443±0.259 |
| 11 | 8/M | 0.144±0.171 (20) | -0.030±0.290 (11) | 0.051 | 0.016 | 0.79 | 1.09 | 1.37 | 0.40 | 0.50 | 0.235±0.271 |
| 12 | 7/F | 0.173±0.147 (17) | -0.227±0.383 (19) | <0.001 | <0.001 | 1.35 | 0.96 | 0.71 | 0.84 | 0.63 | 0.756±0.350 |
| 13 | 7/F | 0.126±0.178 (21) | -0.123±0.212 (25) | <0.001 | <0.001 | 1.26 | 0.85 | 0.67 | 0.64 | 0.51 | 0.289±0.256 |
| 14 | 10/M | 0.370±0.206 (22) | -0.267±0.270 (23) | <0.001 | <0.001 | 2.64 | 0.85 | 0.32 | 0.77 | 0.29 | 0.297±0.249 |
| **Avg** | | 0.205±0.152 (19.1) | -0.197±0.199 (19.3) | | | 2.45 | 0.94 | 0.50 | 1.00 | 0.44 | 0.419±0.243 |

Δx: Horizontal eye displacement in 80–160 ms time window (deg); N: number of correct trials; **p**: unpaired t-test; **p$_{NP}$**: Mann-Whitney U test; **d**: Cohen's d; **min d**: Cohen's d from power test; **min d$_{NP}$**: Cohen's d from bootstrap-based power test.

signals that the ones we measured. The ratio between the minimum *d* that could have been reliably detected and the actual *d* tells us how much smaller a signal we could have detected (assuming that, as in well trained subjects, the variance in the data is not a function of the strength of the signal). We found that, in almost all subjects, eye displacements of one half to one third of the ones we recorded could have been detected.

We further found that vertical OFRs (difference between upward and downward displacements) were slightly, but not significantly (paired t-test, p = 0.687; Wilcoxon signed rank test,

**Table 2. Responses to vertically drifting images.**

| Subj | Age/Sex | Δy±SD(N) UP | Δy±SD(N) DW | p | p$_{NP}$ | d | min d | min d/d | min d$_{NP}$ | min d$_{NP}$/d | OFR_V±SD |
|------|---------|-------------|-------------|---|-----|---|-------|---------|---------|-----------|----------|
| 1 | 13/M | 0.355±0.052 (17) | -0.320±0.092 (16) | <0.001 | <0.001 | 9.12 | 1.01 | 0.11 | 4.56 | 0.50 | 0.675±0.106 |
| 2 | 8/M | 0.089±0.148 (17) | -0.122±0.237 (17) | 0.005 | <0.001 | 1.07 | 0.99 | 0.93 | 0.51 | 0.48 | 0.211±0.279 |
| 3 | 8/F | 0.294±0.282 (13) | -0.236±0.188 (16) | <0.001 | <0.001 | 2.26 | 1.09 | 0.48 | 1.12 | 0.49 | 0.531±0.339 |
| 4 | 7/M | 0.194±0.164 (16) | -0.309±0.146 (18) | <0.001 | <0.001 | 3.26 | 0.99 | 0.30 | 1.16 | 0.36 | 0.503±0.219 |
| 5 | 12/M | 0.213±0.168 (18) | -0.313±0.180 (15) | <0.001 | <0.001 | 3.03 | 1.01 | 0.33 | 0.87 | 0.29 | 0.526±0.246 |
| 6 | 12/M | 0.178±0.117 (22) | -0.139±0.115 (24) | <0.001 | <0.001 | 2.73 | 0.85 | 0.31 | 0.72 | 0.26 | 0.316±0.164 |
| 7 | 13/F | 0.215±0.123 (20) | -0.301±0.155 (23) | <0.001 | <0.001 | 3.66 | 0.88 | 0.24 | 1.83 | 0.50 | 0.515±0.198 |
| 8 | 13/M | 0.188±0.285 (23) | -0.181±0.139 (23) | <0.001 | <0.001 | 1.65 | 0.84 | 0.51 | 0.62 | 0.38 | 0.369±0.317 |
| 9 | 12/M | 0.041±0.131 (21) | -0.164±0.079 (21) | <0.001 | <0.001 | 1.90 | 0.89 | 0.47 | 0.59 | 0.31 | 0.205±0.153 |
| 10 | 6/F | 0.237±0.129 (21) | -0.206±0.225 (22) | <0.001 | <0.001 | 2.40 | 0.88 | 0.36 | 0.87 | 0.36 | 0.443±0.259 |
| 11 | 8/M | 0.112±0.176 (20) | -0.123±0.206 (18) | 0.001 | <0.001 | 1.23 | 0.94 | 0.76 | 0.57 | 0.46 | 0.235±0.271 |
| 12 | 7/F | 0.244±0.243 (19) | -0.512±0.252 (18) | <0.001 | <0.001 | 3.05 | 0.95 | 0.31 | 0.92 | 0.30 | 0.756±0.350 |
| 13 | 7/F | 0.165±0.212 (21) | -0.124±0.143 (20) | <0.001 | <0.001 | 1.59 | 0.90 | 0.57 | 0.77 | 0.49 | 0.289±0.256 |
| 14 | 10/M | 0.234±0.172 (19) | -0.064±0.181 (21) | <0.001 | <0.001 | 1.68 | 0.91 | 0.54 | 0.99 | 0.59 | 0.297±0.249 |
| **Avg** | | 0.197±0.172 (19.1) | -0.222±0.167 (19.4) | | | 2.76 | 0.94 | 0.44 | 1.15 | 0.41 | 0.419±0.243 |

Δy: Vertical eye displacement in 80–160 ms time window (deg); N: number of correct trials; **p**: unpaired t-test; **p$_{NP}$**: Mann-Whitney U test; **d**: Cohen's d; **min d**: Cohen's d from power test; **min d$_{NP}$**: Cohen's d from bootstrap-based power test.

p = 0.502), larger than horizontal (difference between rightward and leftward displacements) ones. Horizontal and vertical OFRs (Fig 5A) were positively and significantly correlated (Pearson's r = 0.53, p = 0.0503; Spearman's r = 0.60, p = 0.023). There were however six cases in which one was significantly larger than the other (red dots in Fig 5A), even considerably so, indicating that there can be large directional variability in OFRs in individual subjects, something that had been previously noted in a smaller group of research subjects (Gellman et al., 1990). We also found that neither horizontal (Fig 5B, Pearson's r = 0.09, p = 0.762; Spearman's r = 0.02, p = 0.945) nor vertical (Fig 5C, Pearson's r = 0.01, p = 0.963; Spearman's r = 0.06, p = 0.837) OFRs were significantly correlated with age. Similarly, the effect size (Cohen's *d*) was not significantly correlated with age for neither horizontal (Pearson's r = 0.26, p = 0.375; Spearman's r = 0.19, p = 0.505) nor vertical (Pearson's r = 0.39, p = 0.165; Spearman's r = 0.23, p = 0.428) motion. It thus appears that the ocular following system is well developed at least by age six.

## Discussion

We showed that it is feasible to detect OFRs in children, using a non-invasive video-based eye tracking system, in a clinical setting, with minimally trained and instructed subjects, and in a single, short, recording session (less than 3 minutes). While recording in pre-verbal children and infants will undoubtedly require overcoming additional obstacles (above all keeping the subjects "engaged" and dealing with more limited head stabilization), our results indicate that it is an effort worth pursuing.

The stimuli we used here were selected because they are known to induce robust OFRs in adults [12]. In most clinical or developmental applications, comparisons between the responses induced by different stimuli will need to be made, and hence smaller differences will likely need to be detected. Our power analysis indicates that, on average, 15 to 20 valid trials should be enough to detect signals of half the magnitude we recorded in most subjects; detecting anything considerably smaller would, however, require either a longer recording session or multiple recording sessions, the approach routinely taken in a research setting. With young children in a clinical setting this would be difficult, and might impose a hard limit on the value

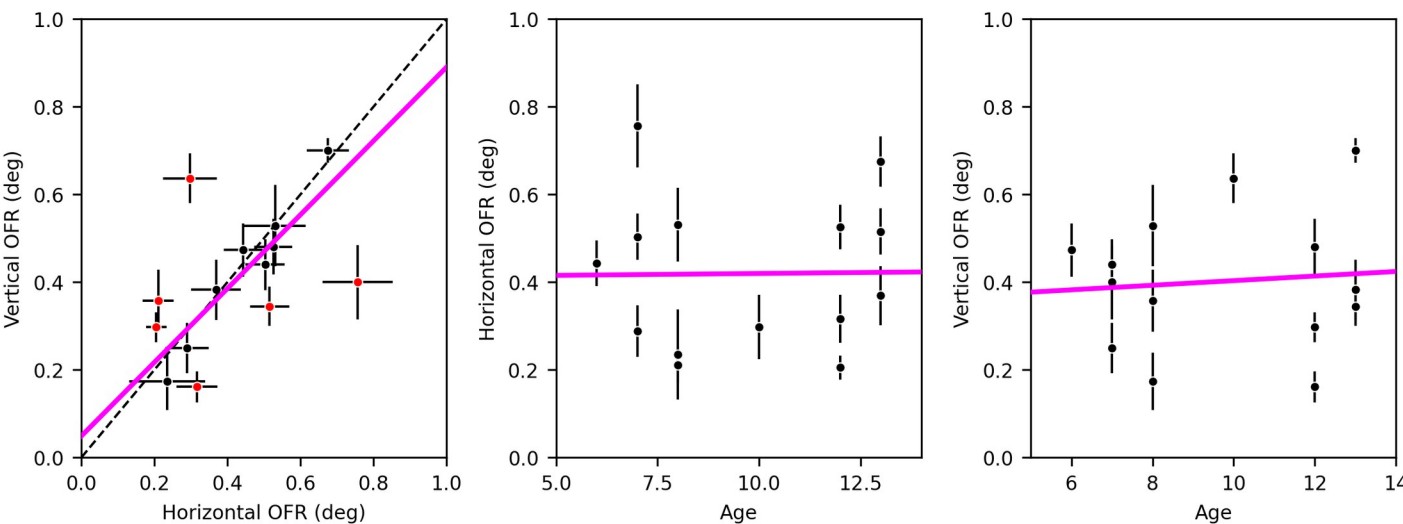

**Fig 5. A)** Relationship between the magnitude of horizontal and vertical OFRs (mean ±1 SEM bars) with trend line (magenta, type II regression). Square gray symbols indicate that the difference between the two is significant (p<0.05, based on non-parametric, bootstrap based, analysis). **B)** The magnitude of horizontal OFRs is not correlated with the age of the subject. **C)** The magnitude of vertical OFRs is not correlated with the age of the subject.

of OFRs in this context. A further point to consider when recording in younger children is that, in selecting our recording time points, we have assumed that the latency of ocular following responses is similar to that of adults. For other types of eye movements, such as saccades, vergence and smooth pursuit, it has been found that latency decreases with age in children [e.g., 20–23]. If the latency of OFRs were also to decrease with age, using a measuring window suited for adults might result in underestimating the magnitude of responses, and possibly missing them altogether. This was not an issue in our study, indicating that in the age group we studied the latency of ocular following responses is not much larger than 80 ms, and certainly well short of 160 ms, but it might become an issue in younger cohorts. A study of the dependency of ocular following latency on age, requiring a higher temporal resolution than the one used here, would thus be useful to guide studies in younger patient populations.

Our data suggest that more precise eye movement recordings would not lead to significant improvements, as the SD of the data is, at around 0.2˚, more than 5 times the resolution of our eye tracking system. Even recording with eye coils would thus result in only a negligible increase in effect size. An angle of 0.2˚ is thus also a good reference to evaluate the suitability of any video-based recording system to the task at hand: To minimally degrade the small OFRs, any recording system should ideally have a resolution (measured in the context of an OFR recording experiment) on the order of 0.1˚ or less. Much of the variability we observed is probably related to what could be called "unsteady fixation". What is commonly observed in recording ocular following in adults is that with time subjects become better at fixating, suppressing microsaccades, so that experienced subjects have a standard deviation of responses that is up to 3–4 times smaller than that observed in our subjects [8, 24]. Steadier fixation also results in much fewer trials having to be discarded in experienced subjects. Unfortunately, in a clinical setting, it is not practical to provide subjects with enough experience. Furthermore, it has been shown that fixation ability improves with age, and it is immature in young children [25, 26]. Thus, this increased level of variability in ocular following responses compared to adult research subjects must be factored in.

We found no significant correlation between subject age and amplitude of movements, at least in the tested range (Fig 5), suggesting that the ocular following system is well developed by age six, and is likely to be present in younger children. This is in line with studies in monkeys and with the development time course of brain areas that are known to be involved in OFR generation in adults. In monkeys, large-scale stimuli moving rapidly are detected as early as 10 days post-birth [27, 28]. In humans, global motion is processed as early as 2 months of age [29–31], indicating that the middle temporal area (MT) and possibly the medial superior temporal area (MST), the neural substrates of global motion perception and OFRs [3, 32, 33] develop very early.

Other types of eye movements also develop quite early. Saccades are already present at one month of age [34] (although their latency is initially large, and decreases up until at least 12 years of age [20, 22, 23, 35, 36]), smooth pursuit can be elicited by two months [37, 38], although it also requires time to fully mature, with its gain increasing quickly in the first few months [38] and then more slowly up to age 15 at least [23, 39]. The vestibulo-ocular reflex is present almost from birth and the opto-kinetic nystagmus by 6 weeks of age [40], but they are mediated mostly by subcortical pathways.

Although it is not currently known at what age OFRs emerge, our findings support the hypothesis that they develop early. This is important because, obviously, the ocular following system needs to be itself developed before it can be used to diagnose other developmental issues. Our primary area of interest is the development of stereovision, but is should in principle be possible to use OFRs to probe the development of the dorsal visual pathway in general [3, 32]. For example, a complete lack of OFRs might indicate a maldevelopment of motion

detection mechanisms in area MT. Deficits in smooth pursuit responses would certainly be more easily detected, but because of the involvement of frontal areas in controlling smooth pursuit they might be a less selective symptom compared to OFR deficits.

In conclusion, our results demonstrate that the ocular following system is well developed by age six, and that recording OFRs in young children is feasible. Almost forty years after their discovery, OFRs might yet become a valuable tool for tracking visual development and for early diagnosis of maldevelopment, such as stereoblindness/amblyopia [11]. Given their minimal need for subject cooperation they might represent a useful tool for the study of visual function in other patient groups that are often difficult to study, such as the elderly and subjects with developmental or psychiatric disorders.

## Acknowledgments

The authors would like to thank orthoptist Rita Silvia Giganti and Francesca Malannino for their support during the data acquisition, Dr. Boris Sheliga for providing the data for Fig 1, and two anonymous reviewers for their helpful comments.

## Author Contributions

**Conceptualization:** Christian Quaia, Bruce G. Cumming, Stefano Pensiero, Agostino Accardo.

**Data curation:** Aleksandar Miladinović, Christian Quaia, Miloš Ajčević, Laura Diplotti, Stefano Pensiero, Agostino Accardo.

**Formal analysis:** Aleksandar Miladinović, Stefano Pensiero.

**Funding acquisition:** Bruce G. Cumming, Stefano Pensiero, Agostino Accardo.

**Investigation:** Aleksandar Miladinović, Christian Quaia, Miloš Ajčević, Laura Diplotti, Bruce G. Cumming, Stefano Pensiero.

**Methodology:** Aleksandar Miladinović, Christian Quaia, Laura Diplotti, Bruce G. Cumming, Stefano Pensiero, Agostino Accardo.

**Project administration:** Stefano Pensiero.

**Resources:** Bruce G. Cumming, Stefano Pensiero, Agostino Accardo.

**Software:** Aleksandar Miladinović, Christian Quaia.

**Supervision:** Christian Quaia, Miloš Ajčević, Laura Diplotti, Stefano Pensiero, Agostino Accardo.

**Validation:** Aleksandar Miladinović, Christian Quaia, Miloš Ajčević, Laura Diplotti, Stefano Pensiero.

**Writing – original draft:** Aleksandar Miladinović, Christian Quaia, Stefano Pensiero.

**Writing – review & editing:** Christian Quaia, Miloš Ajčević, Laura Diplotti, Bruce G. Cumming, Stefano Pensiero, Agostino Accardo.

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
