## [Decision Letter · Decision Letter 0]

18 Aug 2022

PONE-D-22-19294Ocular-following responses in school-age childrenPLOS ONE

Dear Dr. MILADINOVIĆ,

Thank you for submitting your manuscript to PLOS ONE. After careful consideration, we feel that it has merit but does not fully meet PLOS ONE’s publication criteria as it currently stands. Therefore, we invite you to submit a revised version of the manuscript that addresses the points raised during the review process.

We look forward to receiving your revised manuscript.

Kind regards,

Nick Fogt

Academic Editor

PLOS ONE

Journal Requirements:

Additional Editor Comments:

Both reviewers believe that there is potential in this paper.

However, there are a number of comments from both reviewers that should be addressed (reviewer #1 included a Word file with comments) in a revision.

In particular, the authors should carefully consider the additional (basic) methodological details that reviewer #2 has requested. Reviewer #1 has suggested that the authors consider a methods paper just covering the development of the method. A reasonable compromise would be that the authors move any details regarding the development and assessment/validation of the equipment to an appendix. That would help to "clean up" some of the text that reviewer #1 refers to. Some of the methodological details that reviewer #2 has requested can likely go in that appendix as well.

Reviewers' comments:

Reviewer's Responses to Questions

**Comments to the Author**

1. Is the manuscript technically sound, and do the data support the conclusions?

Reviewer #1: Partly

Reviewer #2: Yes

2. Has the statistical analysis been performed appropriately and rigorously? 

Reviewer #1: Yes

Reviewer #2: Yes

3. Have the authors made all data underlying the findings in their manuscript fully available?

Reviewer #1: Yes

Reviewer #2: Yes

4. Is the manuscript presented in an intelligible fashion and written in standard English?

Reviewer #1: Yes

Reviewer #2: Yes

5. Review Comments to the Author

Reviewer #1: Please see my attached document for my comments to the authors. This attachment includes line-specific notes and questions/concerns regarding the work as well as typgraphical and grammatical corrections. Thank you!

Reviewer #2: The study demonstrates that short-latency ocular following responses can be assessed, even crudely, in school-age infant (6-12 years old) using a custom-made, low rate video eyetracker and short (3min) recording sessions (about 20 trials/condition). Under these rather stringent conditions, the authors show that direction-selective (U/D/L/R) initial responses (60-120ms after stimulus onset) can be accurately estimated from only 3 frames. Small responses (in the order to 1/4 of a degree of visual angle) are measured reliably across 12 participants. There is no effect of age upon response amplitude, meaning that by the age of 6, reflexive tracking eye movements are mature.

This approach open the doors for future large-scale clinical studies as reflexive, short-latency ocular responses are a reliable proxy of low level vision (motion processing, binocular vision).

I have only minor comments/suggestions:

1) The introduction is written for experts in eye movements. For instance (line 4-5), the fact that responses are small is only understandable if ones know that these responses are recorded over only a few hundreds msec (typically 200-300ms after stimulus onset) and therefore amplitude are estimated during the initial, open-loop, accelerating phase of the tracking responses. It doesn't mean that the eye will not reach later on the target velocity (or a significant fraction of it). Moreover, a stimulus input of 50°/s is rather high. Again, it sounds classical for OFR afficionados but may be seen as not relevant for non expert. I would recommend to somehow lengthen the introduction to better explain the key properties of ocular following (not only mentioning review papers) and the very specific conditions (open-loop phase, short stimulus duration, unpredictable visual conditions....) under which these very small, but highly reliable responses can be measured.

2) Methods: In the same vein, it will sound very bizarre to non experts why recording only 3 data points are sufficient. Again, it works because of the very reproducible eye velocity profile of ocular following that can be summarized to two linear epochs: a pre-response flat line (0-80ms) and a linear ramp at nearly constant speed (80-160ms). Under those properties, it is sufficient to acquire only 3 frames. May be it would be worth to illustrate in Figure 1, some exemples of eye velocity profiles (and their parametric dependencies) of already published work by the same group. In fact, most ocular following studies use the change in eye position over the same time windows even when using much more sophisticated eye movement recording techniques, a measure that is exactly what the current study samples directly. Thus, non experts would clearly understand why this approach is both safe (limiting the IR illumination), sufficient and consistent with empirical, lab-based studies.

3) Results: At the individual trial level, there seems to be a significant fraction of cross-talk between horizontal and vertical eye movements (e.g. Figure 2). Such orthogonal biases appear to canced out across trials. I have two questions: (i) is this cross-talk commensurate with what is classicaly observed using other techniques or is it specific to your video-eyetracker ? (ii) Can you give a statistical estimate of such dispersion along the orthogonal direction ?

4) Discussion: The authors pinpoint the technical interests (low IR illumination, small trials repetition..) for clinical studies. I would suggest that they discuss further to potential uses beyond visual motion developmental studies.

First, would this approach be usable for short-latency vergence responses ? Investigating stereoblindness is large populations with eye movements is difficult. Under which condition the current approach would be useful ? For instance, without getting to the trouble of stereo vision, vergence responses driven by optical flow pattern would be very interesting to probe. Second, how this apparatus would be compatible with stereo vision systems ?

Second, gathering enough data is clinical setting is painful not only for children but also for all sorts of patients (just to mention a few: elderly, psychiatric/neurological debilitating diseases, young adults with severe developmental disorders...). Being able to quickly and reliably estimate the sensory processing capabilities and their variabilities would be a plus for many developmental, neurological and ophthalmological studies.

6. PLOS authors have the option to publish the peer review history of their article (what does this mean?). If published, this will include your full peer review and any attached files.

Reviewer #1: No

Reviewer #2: No

---

## [Author Response · Author response to Decision Letter 0]

13 Sep 2022

We thank the reviewers for their thorough and thoughtful comments. We have incorporated their suggestions, and tweaked the text and figures to improve the quality of the presentation. We believe that this has resulted in a considerably clearer presentation of our results, more easily accessible to a wider audience. A point-by-point reply to their comments follows.

Reviewer #1

Abstract

Line 22: Recently, it has been…

Line 23: and, thus, might have…

Lines 31-32: … recording sessions. Although, we found… (And, welcome to working with kids – A LOT of

variability! ;) )

Fixed. 

Regarding the variability, we are not sure that it is specific to children. At the NEI, where OFRs have been recorded (in adults, with scleral coils) for decades, a wide variability in the magnitude of OFRs across subjects has been observed (see Figure 1, added to show sample OFRs recorded in adults with eye coils at NEI), but since a small number of subjects was used in each experiment, this variability has never been able properly quantified. Because all the tuning properties studied were always highly consistent across subjects, the field has assumed that these differences in magnitude arise at the motor end of the circuit, but there is no strong evidence to support this conclusion.

Introduction

Line 48: humans and non-human primates…

(Line 62: random thought – also, maybe, non-verbal school-aged kids and adults? Added to the

discussion?)

Line 66: write out these numbers?

Fixed

Lines 70-81: These lines seem to read more like an abstract than a general purpose statement to setup

the focus of the study/paper. As for the purpose: to be the first to search for/record OFR’s in children, to

determine if/when these movements are developed,…?

We shortened this section, and refocused it as suggested.

Lines 70-72: Returning after reading the methods… Revisiting the whole of this work, it seems to have

dual purpose: (1) to develop a custom, video eye-tracking device for use with children in a clinical

setting, as well as to verify its validity compared to a gold standard such as a search coil; and (2) attempt

to measure and document OFRs in children aged 6-14. The first of these may be its own paper? Maybe

not, yet it seems to have led to an addition of introductory information and results mixed into the

Methods section.

The video-based system was developed by recording OFRs in adults, while simultaneously recording (in some sessions) the same eye with a scleral eye coil. Once we verified that it performed reasonably well in adults, we started using it to record in children. Because we haven’t published a separate paper on the method itself, we are describing it at some length in the Methods section, which we have now cleaned up to rectify the problems highlighted by the reviewer.

Methods

Line 84: “Sixteen healthy…”; please write out this number, especially being at the start of the section.

Fixed

Lines 89-90: What was the timing between the complete exam and the research visit? The same say was

not likely if they were dilated during the exam? If returning for a second visit, how much time had past?

This could be important if subjects where prescribed a new/updated spectacle prescription and needing

an adaption period to the new glasses.

Ophthalmological exams and the OFRs recordings were performed on the same day. Each child was first examined by an ophthalmologist to determine visual acuity through non-cycloplegic refraction; the parents of those that met our criteria were then approached to receive informed consent; in those that agreed, the short recording session was then performed; finally, the child was seen again by the ophthalmologist for a cycloplegic refraction test (and usually other tests). Our plan was to subsequently exclude from the study, without analyzing their data, children with a significant difference between the two refraction tests, but there were none.

Line 96: A reference to the setup shown in Figure 1a?

Line 102: “(Figure 1b).”

Line 109: “… composed of four randomly-intermixed trials…”

Fixed

Lines 117-122: I like this addition! ☺

Thanks! The experiment itself is objectively boring, so we tried to come up with a way of keeping them motivated to fixate and avoid blinking. 

Lines 123-181: This seems to be a separate, but related!, study. Please refer to my comments for the

purpose statement at the end of the Introduction. The Hopf paper from 2018 (doi:

10.1371/journal.pone.0204008) might be of assistance in terms of the organization/format for what I

am recommending. It seems distracting to have additional information introduced and results reported

within the Methods section.

See above. We have considerably edited this section to address the issue.

Line 144: “…(Figure 1a).”

Fixed

Lines 144-146: I can see the authors’ argument for the three frames of data collection, vs continuous

data, yet is using 80 ms for the latency appropriate for use with younger children? Other eye movement

studies, including the Hopf paper mentioned above, have shown that most eye movement dynamics are

adult-like with the exception of latency periods decreasing into the pre-teen/teenage years. Therefore,

would half of your study population have a longer latency period and, then, require a different framing

rate to measure position/displacement?

It’s a fair criticism. We think that a direct comparison with the latency of voluntary movements (such as saccades and smooth pursuit) is probably not appropriate, given that such movements are affected by attentional/frontal cortex control to a much larger extent than the reflexive movements that we are studying here. Nevertheless, it is possible that the window we have used is not ideal (i.e., if the latency of OFRs in children were, say, 100ms, we should have recorded at times 100 ms and 200 ms, and by recording at 80 ms and 160 ms we would be measuring the eye displacement over the initial 60% of the open loop period). Since we reliably detected OFRs in all subjects this was not a limiting factor in our study, but in recording younger children it might become so. With our current setup we can easily record a frame every 20 ms, and down to 5 ms if we reduce the size of the image being stored to disk. In a future study we will thus be able to better estimate the latency of the responses, as a function of age. We still believe that using only three frames is the best approach, as it allows to minimize IR illumination, but knowing which time window to use to maximize the signal magnitude will certainly be helpful. We now note this in the discussion.

Lines 168-171: I appreciate that the authors calibrated the system to a standard measure to validate

their protocol. (Again, maybe a separate purpose/paper?) Were both the two subjects adults? Authors?

If so, could this be done with an older child? Many children are very successful with contact lens wear so

finding one or two 10-12-year-olds to wear a search coil may be possible? (All, of course, pending IRB

approval.)

The subjects on which the system was developed (at NEI) and tested were adults, one an author (CQ) and the other not. The scleral eye coil is not a contact lens, it is a rather thick and hard silicon annulus with an embedded coiled wire, which then exits the annulus at the inner canthus. For subjects that are not used to it, it is quite uncomfortable to wear, especially in the first one-two recording sessions. For 20% of subjects it is a one-time experience that they do not wish to repeat. While technically possible, it would thus be quite difficult to collect reliable data in children. Furthermore, since our software simply fits an ellipse to the pupil and computes its displacement across frames, it is hard to see how the process could differ in children and adults, and it would thus be very difficult to make a good case for it to any IRB.

Line 177-178: If this is true, you have the data to do so? An argument to first publish this as a smaller

paper first vs the dual purpose option?

We do plan to publish a methods paper on the device, but we are not yet ready to do so, as we are in the process of making the software more user friendly, to be able to make the software (as well as all the technical and electrical drawings for hardware parts) freely available to researchers interested in pursuing this line of research. We feel, however, that the current paper can stand on its own, given the rather extensive description of the method we have provided. Since our paper might get other researchers interested in measuring OFRs in children (possibly using other techniques or devices), we think that it would not be beneficial to the field to delay the publication of these results.

Lines 194-197: Looking at the Hellstrom paper (1997 vs 2009?), this is an axial length measure? Wouldn’t AL be easy to measure on your subjects to verify an average used? As mentioned, a small difference would not matter, yet does your population fits this report from the literature, especially if comparing a 6 yo with a 13 yo? Italians vs Swedes? I would predict that you would have more variability with your older subjects given the increase in the prevalence of myopia over the past 25 years. If the author’s calculations can show that a range of ALs do not significantly change the degrees/pixel, the average used is reasonable. However, the reference provided seems dated when there is a TON of current literature on axial length in school-aged myopes, including controls.

Thanks for picking out the 1997/2009 issue. On the Wiley website the paper is listed as first published in 2009, and our reference management software must have picked up that date. We fixed the reference. 

We could have recorded the axial length, but we felt that it was not necessary, since from the literature it seemed to be rather constant in the age range we planned to test in this first study. As noted in the Methods, we selected subjects with minimal visual defects, and only three were mildly myopic (less than one diopter). Differences across subjects can thus be expected to be of the order of ±10%, as in Hellstrom’s paper, which could account for only a small fraction of the 4-fold variability in the magnitude of responses across subjects, shown in Figure 4. In younger subjects this might become a more important issue, which we now note in the Methods (although in ocular following experiments all conclusions are typically based on the ratio between responses to two different stimuli, not on the absolute size of responses).

Line 202: The head was stabilized (Line 92) during testing so large head movements should have been

limited?

“Should” was well chosen. We used a head band to keep the forehead in contact with the head rest, but we did not feel comfortable making the head band as tight as we normally do in adults, so it acted more as an encouragement/reminder to keep the head stable than as enforcing stability. In younger children and infants we will not even have the luxury of having such limited stabilization, and thus we focused mostly on trying to maximize the comfort of the children (and their closely observing parents).

Line 209: “… the middle 68th percentile of these values,…”

Fixed.

Lines 209-210: I like the use of the mid-68th percentiles. Maybe a table to show the amount (number or

percent) of trials dropped for each subject? Or, a column for reliability added to Tables 1 & 2?

The number of valid trials for each condition (out of 25 presentations) for each subject is already indicated in the tables (in parenthesis, after the displacement value). It was not indicated in the table caption, and so it went unnoticed. We now indicate it. 

Lines 210-212: This belongs in the results section?

Whether it belongs in the Methods or in the Results is debatable. Because excluded trials are not analyzed, we think that it’s more of a Methods than a Results point, and so we’d be inclined to leave it where it is, but if the reviewer feels strongly about it we could move it in the Methods section.

Lines 245-254: This section is typically listed at the beginning of a methods section? Also, what about the

two subjects used to calibrate/verify the eye tracking device? Or, were these two authors?

We moved the section to the beginning of the Methods section. The subjects whose eye movements (not presented here) were used to develop the measuring system were covered by a different protocol, at the National Eye Institute, NIH.

Results

Line 258: “In Figure 2,…”

Line 260: “(Subject 10).”

Fixed

Lines 260-262: Please see my previous concern regarding the use of these timing frames.

See above, and added text in the discussion.

Line 270: “Note, however, the large…”

Line 277: “In Figure 3, scatter plots…”

Line 293: On the top row of Table 1, the third and fourth columns should be rightward and leftward?

Line 295: Change in x? Horizontal eye displacement?

Fixed

Lines 300-315: Some of this seems to include a discussion of the results and should be moved. Please

make a simple statement of the results here.

We kept the first paragraph, since we feel that it is important to reassure the reader that the measure we are reporting is appropriate. Some might find it weird that we are reporting a change in position for a tracking eye movement, and suspect that many readers might skip the Methods section, where this is also addressed. We have deleted the other two paragraphs, and made those points more succinctly in the Discussion.

Lines 300-304: Would it first be beneficial to runs trials with continuous measures to plot the

instantaneous velocities for children and estimate their latencies as a function of age? (Please see my

previous notes on latencies of eye movement dynamics in children, Lines 144-146. This could potentially

be another goal/measure to add to the study or another separate paper?) If the latency is different from

that of adults, at least for the younger children, the, would you need to use different time intervals? The

quantification based on use of time displacements seems an adequate method for OFR measures

provided that these latencies are appropriate.

See above. What we have demonstrated is that, in this age group, this time window is adequate, although it might not be optimal, a point that we now make in the Discussion.

Line 315: “…this variability…”; may need to define if moving/updating preceding paragraphs.

Fixed.

Line 339: “intact” vs “fully developed?” More on this in my overall comments.

We are concerned that the term “intact” might be open to multiple interpretations. However, we see the point being made by the reviewer, and “fully developed” might indeed be too strong of a statement (especially given that we do not have the data to show that the latency is stable). We thus decided to settle for “well developed”. Hopefully the reviewer will found it to be an acceptable description.

Discussion

Lines 347-349: Yes – agreed!

Lines 349-351: Something for future study, I do think that maintaining fixation in infants will be easier

than you think; please see literature on Teller Acuity.

That is welcome news!

Lines 352-354: Again, the OFR is “intact” by age six years. I would like to see further discussion of how

this might compare to other types of eye movements and the characteristics in children, especially as

most dynamics are adult-like. (The exception is with latency.)

Lines 354-362: I like where the authors are leading to here. I’d like to see this more developed into

further discussion. Other eye movements (fixations, saccades, pursuits, vergence) are seen in infancy.

Then, specifically, how can this development of OFR’s be used to diagnose other developmental issues?

We have expanded this section of the Discussion, and we have moved it towards the end. We believe we have addressed the points raised by the reviewer, although we have tried to do it as succinctly as possible, to avoid speculating too much.

Line 361: “… important because, obviously, the ocular…”

Lines 363-364: “… induce robust OFRs in adults [13].” Also, this first sentence of the paragraph seems to be displaced.

Line 369: “15 to 20,” or “15 out of 20?”

Line 370: “… smaller would, however, require either…”

Fixed

Lines 374-377: Good! Maybe consider verifying with an older child? (Many kids are successfully wearing

CLs by 10-12 years of age so a search coil for 1-2 subjects would be reasonable? If IRB approved, of

course.)

See above.

Line 377: “An angle of 0.2⁰ is thus…”

Line 382: “Forty years…”

Line 383: “… tool for tracking visual development…”

Fixed

References

Lines 433-5: I looked up this article out of interest and a question on Lines 194-197. Volume 75 was

published in 1997, not 2009 (https://pubmed.ncbi.nlm.nih.gov/9374253/). The authors may need to

further proofread/verify the references.

Fixed, see above.

General Comments

Overall, I agree that the authors have presented a well-written argument that OFRs are present in

children. They have successfully developed and verified an eye tracking device to measure such OFRs. I

still seem to be stuck on the latency of OFRs in children compared to adults. Given the short latency of

these eye movements, it may not be significant, yet it seems to be a gap in the literature -- especially as

other types of eye movements (fixations, saccades, pursuits, vergence) are present in infancy yet have

latencies that tend to decrease into puberty. It may be beneficial to try to measure OFR latency in

children first. (If different from the 80ms used, the measured displacements will be within different

framing measures and inaccurate. And, Gellman (ref #13) and Leigh & Zee report 70ms in adult humans.)

The authors likely have appropriate means to be able to do this. This could mean that the study could be

further strengthened by proposing a three-part goal:

(1) to develop and verify an eye tracking device for clinical measurements of OFRs in children,

(2) (2) to estimate the latencies of of OFRs in children, and then,

(3) (3) to measure horizontal and vertical displacements of OFRs in children, comparing them to

those of adults.

These final measurements could then be used for detection of other visual developmental delays such

as strabismus and amblyopia.

Articles on eye movement dynamics with age, including latency periods

(just a start, but what I found in less than a few minutes)

Quan, 2002 (PMID: 12202513)

Sinno, 2020 (DOI https://doi.org/10.3766/jaaa.19049)

Irving, 2006 doi:https://doi.org/10.1167/iovs.05-1311)

Salman, 2006 (https://doi.org/10.1016/j.visres.2005.06.011)

Not having a good handle on the latency is certainly less than ideal, but we do not see it as a fatal flaw; it is simply something else that will need to be investigated, and it might very well be important in younger children. If we had reported a negative result, i.e., inability to detect OFRs, it would have been certainly fair to say that having chosen an incorrect time window could be responsible for the results, making them uninterpretable. However, since we identified significant OFRs in all directions in all subjects, at most one could say that the OFRs would have been even larger had we selected an optimal time window (assuming, and it is not a given, that we have not in fact used the optimal window). Our measures are thus in some sense conservative estimates of the OFRs magnitudes that one could expect.

The papers indicated by the reviewer, as well as others in the literature, deal with the latency of saccades (in same cases disconjugate, although usually conjugate) or with the gain of pursuit. As we noted above, these are movements that have much longer latencies than OFRs (in adults, ~180-220 ms vs ~70-80 ms), and are well known to be controlled, especially in their onset, by frontal cortical areas, which take many years to fully develop. The reflexive eye movements we are studying here have no room for such frontal mediation: in monkeys one finds latencies as short as 45 ms, which, taking into account that responses in V1 have latencies of 30 ms, leaves only 15 ms for the signal to travel through MT, MST, the pontine nuclei, the brain stem and out to the eye muscles, making them almost certainly the fastest cortically mediated responses. A more appropriate comparison (at least partially so, given its subcortical mediation and the fact that it is present even in cortically blind children) would be with the opto-kinetic nystagmus, but we have not been able to find any latency measure for it in children, as the focus is always on its gain and nasal-temporal asymmetries.

(e.g., Valmaggia et al 2004, http://dx.doi.org/10.1136/bjo.2004.044222)

70 ms is indeed the latency reported by Gellman et al., but OFR latency is affected by the size of the stimulus and the contrast of the image. Gellman used a full screen stimulus (90° x 90°), at 100% contrast (black and white dots, in a completely dark room). We used a more modest size, compatible with the size of our screen and its distance from the subject, and the image contrast was 80%. Under these conditions 80 ms is a reasonable latency (in adults). 

Reviewer #2

The study demonstrates that short-latency ocular following responses can be assessed, even crudely, in school-age infant (6-12 years old) using a custom-made, low rate video eyetracker and short (3min) recording sessions (about 20 trials/condition). Under these rather stringent conditions, the authors show that direction-selective (U/D/L/R) initial responses (60-120ms after stimulus onset) can be accurately estimated from only 3 frames. Small responses (in the order to 1/4 of a degree of visual angle) are measured reliably across 12 participants. There is no effect of age upon response amplitude, meaning that by the age of 6, reflexive tracking eye movements are mature.

This approach open the doors for future large-scale clinical studies as reflexive, short-latency ocular responses are a reliable proxy of low level vision (motion processing, binocular vision)

I have only minor comments/suggestions:

1) The introduction is written for experts in eye movements. For instance (line 4-5), the fact that responses are small is only understandable if ones know that these responses are recorded over only a few hundreds msec (typically 200-300ms after stimulus onset) and therefore amplitude are estimated during the initial, open-loop, accelerating phase of the tracking responses. It doesn't mean that the eye will not reach later on the target velocity (or a significant fraction of it). Moreover, a stimulus input of 50°/s is rather high. Again, it sounds classical for OFR afficionados but may be seen as not relevant for non expert. I would recommend to somehow lengthen the introduction to better explain the key properties of ocular following (not only mentioning review papers) and the very specific conditions (open-loop phase, short stimulus duration, unpredictable visual conditions....) under which these very small, but highly reliable responses can be measured.

We have revised the Introduction, along the lines suggested by the reviewer. It should now be more readable by non-experts.

2) Methods: In the same vein, it will sound very bizarre to non experts why recording only 3 data points are sufficient. Again, it works because of the very reproducible eye velocity profile of ocular following that can be summarized to two linear epochs: a pre-response flat line (0-80ms) and a linear ramp at nearly constant speed (80-160ms). Under those properties, it is sufficient to acquire only 3 frames. May be it would be worth to illustrate in Figure 1, some exemples of eye velocity profiles (and their parametric dependencies) of already published work by the same group. In fact, most ocular following studies use the change in eye position over the same time windows even when using much more sophisticated eye movement recording techniques, a measure that is exactly what the current study samples directly. Thus, non experts would clearly understand why this approach is both safe (limiting the IR illumination), sufficient and consistent with empirical, lab-based studies.

We are pointing out this issue in the Methods section, and then we briefly reiterate it in the Results section, as some readers might skip the Methods and jump directly to the Results. Adding a figure showing some OFR velocity trace recorded with eye coils is a very good idea, especially for readers unfamiliar with these types of eye movements; we have now added it as Figure 1.

3) Results: At the individual trial level, there seems to be a significant fraction of cross-talk between horizontal and vertical eye movements (e.g. Figure 2). Such orthogonal biases appear to canced out across trials. I have two questions: (i) is this cross-talk commensurate with what is classicaly observed using other techniques or is it specific to your video-eyetracker ? (ii) Can you give a statistical estimate of such dispersion along the orthogonal direction ?

We assume that with cross-talk the reviewer here refers to the correlation between horizontal and vertical displacements seen in the sample subject data shown in Figure 2. It is just a fluke correlation arising from using small numbers of trials. When we look across the 112 datasets (14 subjects x 4 directions of motion x 2 epochs), the correlation between horizontal and vertical displacement ranges from -0.78 to +0.78, with a very modest positive bias (overall Pearson’s r=0.068). The eye displacements we measure are usually of the order of one image pixel, rarely exceed two pixels (unless there are sizeable head movements, of course), and changes in pupil size are also very limited (given the short time interval between successive frames). Cross-talk between the horizontal and vertical channel is thus unlikely to arise (unless the camera is slightly tilted, of course), and we did not observe any when we calibrated the system with an artificial eye.

4) Discussion: The authors pinpoint the technical interests (low IR illumination, small trials repetition..) for clinical studies. I would suggest that they discuss further to potential uses beyond visual motion developmental studies.

First, would this approach be usable for short-latency vergence responses ? Investigating stereoblindness is large populations with eye movements is difficult. Under which condition the current approach would be useful ? For instance, without getting to the trouble of stereo vision, vergence responses driven by optical flow pattern would be very interesting to probe. Second, how this apparatus would be compatible with stereo vision systems ?

The approach would in principle be applicable to measuring vergence. Instead of using the nose sticker to track the movements of the head, we would simply track the displacement of the two pupils (which, especially in children, fit within the camera shot with the camera lens we are using), and subtract one from the other. Head movements (as long as they are limited to small translations) would not need to be corrected for. We have not pursued this approach because disparity vergence responses (DVRs), the equivalent of OFRs in the vergence domain, are typically even smaller than OFRs. At NEI they have been recorded for decades (since their discovery in 1996), with two scleral eye coils, but over the years subjects with perfectly normal stereo-vision had to be discarded because their vergence movements were so small that extracting a usable signal would have required averaging over hundreds of repetitions (and, inexplicably, wearing two search coils is more than twice as uncomfortable as wearing one). A solution to this problem presented itself when two of us (CQ, BGC) discovered that OFR responses induced by correlated and anti-correlated patterns are significantly different in stereo-normal, but not in stereo-blind, subjects. This allows to detect stereo-anomalies by recording the movement of only one eye, and we have thus been pursuing this approach.

As far as larger vergence responses (i.e., not DVRs), our setup would be compatible with recording them in stereo vision systems, provided that a view of the eyes is available, possibly by use of half-mirrors. To record OFRs to dichoptic stimuli, our approach is to use an anaglyphic setup with lenses that allow passage of red or blue light, and image the eye that has the “red” lens in front of it (the “blue” lens blocks IR). We wanted to focus this paper on the data, and not make it too much about the recording method. We plan to write up a methods paper, where we will address these issues more fully. 

Second, gathering enough data is clinical setting is painful not only for children but also for all sorts of patients (just to mention a few: elderly, psychiatric/neurological debilitating diseases, young adults with severe developmental disorders...). Being able to quickly and reliably estimate the sensory processing capabilities and their variabilities would be a plus for many developmental, neurological and ophthalmological studies.

That’s a good point. We point that out in the Discussion now.

---

## [Decision Letter · Decision Letter 1]

2 Oct 2022

PONE-D-22-19294R1Ocular-following responses in school-age childrenPLOS ONE

Dear Dr. MILADINOVIĆ,

Thank you for submitting your manuscript to PLOS ONE. After careful consideration, we feel that it has merit but does not fully meet PLOS ONE’s publication criteria as it currently stands. Therefore, we invite you to submit a revised version of the manuscript that addresses the points raised during the review process.

We look forward to receiving your revised manuscript.

Kind regards,

Nick Fogt

Academic Editor

PLOS ONE

Journal Requirements:

Additional Editor Comments :

The authors have satisfactorily addressed the majority of the comments from the reviewers. Only a couple of minor comments from one of the reviewers remain to be addressed:

"I would have liked to [have] seen a little more inclusion criteria included with the study

population. In the previous review, I asked about the timing of the study visit with the

patient exam and potentially prescribed spectacle changes. The authors provided a

good explanation yet this should be included in the text of the paper. The later comment

about axial length and lack of significant myopia could also be included here."

In regard to the reviewer comment about perhaps moving the portion of the methods in which the calibration of the device and protocol is described to an appendix, some readers will likely want to know that this is an important aspect of the paper. Therefore, while moving that material to an appendix does not have to be the approach, please add a sentence to the abstract to indicate that methodological considerations/development of the device is described in the paper.

Reviewers' comments:

Reviewer's Responses to Questions

**Comments to the Author**

1. If the authors have adequately addressed your comments raised in a previous round of review and you feel that this manuscript is now acceptable for publication, you may indicate that here to bypass the “Comments to the Author” section, enter your conflict of interest statement in the “Confidential to Editor” section, and submit your "Accept" recommendation.

Reviewer #1: (No Response)

2. Is the manuscript technically sound, and do the data support the conclusions?

Reviewer #1: Yes

3. Has the statistical analysis been performed appropriately and rigorously? 

Reviewer #1: Yes

4. Have the authors made all data underlying the findings in their manuscript fully available?

Reviewer #1: Yes

5. Is the manuscript presented in an intelligible fashion and written in standard English?

Reviewer #1: Yes

6. Review Comments to the Author

Reviewer #1: Please see the attached document that includes comments for the authors in response to their resubmission.

7. PLOS authors have the option to publish the peer review history of their article (what does this mean?). If published, this will include your full peer review and any attached files.

Reviewer #1: No

---

## [Author Response · Author response to Decision Letter 1]

4 Oct 2022

We thank the editor and reviewer for their suggestions, which we have incorporated. A point-by-point reply to their comments follows.

Reviewer #1

I would have liked to seen a little more inclusion criteria included with the study population. In the previous review, I asked about the timing of the study visit with the patient exam and potentially prescribed spectacle changes. The authors provided a good explanation yet this should be included in the text of the paper. The later comment about axial length and lack of significant myopia could also be included here.

We have expanded the description of the inclusion criteria in the Study population section of the methods, along the lines of our previous response. In the Data analysis section we also reiterated that the assumption we made about eye diameter is justified because of the low refractive error in our population.

The flow of the methods still seems a bit choppy with the detour about the calibration of their device and protocol as it is lengthy and distracts from the original purpose. The authors made a comment expecting readers to skip over a lot of the methods, and this distraction may encourage this. Adding an appendix to the paper may be a more appropriate way to include the development of device so that it does not take away from the flow of the paper.

We realize that it is quite a lot to get through, but we feel that some readers might find it important. Since it is all contained in its own labeled section, it would be easy for uninterested readers to skip just it and focus on the rest of the methods. We find that appendices are often even more disruptive to the flow, especially when articles are read on digital devices, on which skipping a section is easy, but navigating to the end to find the appendix, and then back to where one was, is often cumbersome. We thus try to avoid them.

For what was previously line 209, an error still exists, " ...the middle 68th percentile of these values, ..."

Fixed.

I appreciate the author's attention to concerns raised in the comments from both reviewers. The authors seem well versed in techniques and current literature. They raised many great discussion points from both sets of responses stimulating conversation regarding an explanation of reasoning. Some of the statements could very well be included within the text of the paper itself to clarify the work and make it more accessible to others.

We understand the reviewer’s point. We addressed some of the points raised in the previous round only in the reply letter because we tried to strike a compromise between being fully exhaustive and being concise. We have made a good faith effort to include in the manuscript everything that might be of more general interest, while leaving to the reply letter what might be of more special interest. Although we realize that where the line lies is always somewhat subjective, we believe that there is value in trying to keep the discussion as focused as possible.

One general concern, I am not sure how much experience the authors have working with children. Some of their comments could be interpreted as rather negative when mentioning working with children. Many kids will surprise you and complete tasks better than some adults. (I know what a search coil is and regularly perform Goldman tonometry on 6 year olds, an exam element many adults cannot successfully withstand.) Having done eye tracking with calibration on preschoolers, it is possible. Infants will readily fixate on flashing screens while their parents can hold heads or in airplane position; we do this at a slit lamp during routine eye exams. Additional testing with devices may potentially lengthen a comprehensive exam for a child, yet this is already being done. In the next few years, I see axial length measures becoming standard protocol for any young myope. Eye care professionals already use VEP on infants and pre- or non-verbal children if concerned about developmental issues. (Not a fan of it, but it is commonly performed.) Anyway, I mentioned this not so much a specific need for this article. The authors have potential for future work based on these findings and should not see young subjects as any more challenging than other populations. Kids are also far more fun!

We have extensive experience in recording eye movements in adults and school-age children, but not much in infants. We have heard mixed stories from colleagues that have, but we find the reviewer’s comments encouraging. We thank the reviewer again for the many insightful comments.

---

## [Editor Report · Decision Letter 2]

6 Oct 2022

PONE-D-22-19294R2Ocular-following responses in school-age childrenPLOS ONE

Dear Dr. MILADINOVIĆ,

Thank you for submitting your manuscript to PLOS ONE and for your thorough responses to the reviewer comments. Only one item remains: Please submit a revised version of the manuscript that addresses this final point regarding the abstract and described below.

Please add the following to the abstract, as the readership will likely be interested in knowing that there are many methodological details in the paper: add a sentence to the abstract to indicate that methodological considerations/development of the device is described in the paper. As the abstract reads now, it is not clear that all of the details in the development of the method are presented in the manuscript.

We look forward to receiving your revised manuscript.

Kind regards,

Nick Fogt

Academic Editor

PLOS ONE
---

## [Author Response · Author response to Decision Letter 2]

13 Oct 2022

Editor comment: Thank you for submitting your manuscript to PLOS ONE and for your thorough responses to the reviewer comments. Only one item remains: Please submit a revised version of the manuscript that addresses this final point regarding the abstract and described below.

Please add the following to the abstract, as the readership will likely be interested in knowing that there are many methodological details in the paper: add a sentence to the abstract to indicate that methodological considerations/development of the device is described in the paper. As the abstract reads now, it is not clear that all of the details in the development of the method are presented in the manuscript.

Response: Dear Editor, thank you for your comment. We modified the abstract accordingly and added the sentence regarding the developed custom system used for OFRs recording in our study.

Reviewer comments: n/a

--

The authors would like to thank again the Editor the comment, as well as, reviewers for their previous comments that allowed us to improve the paper.

---

## [Editor Report · Decision Letter 3]

27 Oct 2022

Ocular-following responses in school-age children

PONE-D-22-19294R3

Dear Dr. MILADINOVIĆ,

We’re pleased to inform you that your manuscript has been judged scientifically suitable for publication and will be formally accepted for publication once it meets all outstanding technical requirements.

Kind regards,

Nick Fogt

Academic Editor

PLOS ONE

Additional Editor Comments (optional):

Thank you for addressing all of the reviewer and editor comments.